# CCR7 Mediates Cell Invasion and Migration in Extrahepatic Cholangiocarcinoma by Inducing Epithelial–Mesenchymal Transition

**DOI:** 10.3390/cancers15061878

**Published:** 2023-03-21

**Authors:** Mitsunobu Oba, Yoshitsugu Nakanishi, Tomoko Mitsuhashi, Katsunori Sasaki, Kanako C. Hatanaka, Masako Sasaki, Ayae Nange, Asami Okumura, Mariko Hayashi, Yusuke Yoshida, Takeo Nitta, Takashi Ueno, Toru Yamada, Masato Ono, Shota Kuwabara, Keisuke Okamura, Takahiro Tsuchikawa, Toru Nakamura, Takehiro Noji, Toshimichi Asano, Kimitaka Tanaka, Kiyoshi Takayama, Yutaka Hatanaka, Satoshi Hirano

**Affiliations:** 1Department of Gastroenterological Surgery II, Hokkaido University Faculty of Medicine, Sapporo 060-8638, Japan; 2Department of Surgical Pathology, Hokkaido University Hospital, Sapporo 060-8648, Japan; 3Research Division of Genome Companion Diagnostics, Hokkaido University Hospital, Sapporo 060-8648, Japan; 4Center for Development of Advanced Diagnostics (C-DAD), Hokkaido University Hospital, Sapporo 060-8648, Japan; 5NB Health Laboratory Co. Ltd., Sapporo 001-0021, Japan

**Keywords:** CCR7, CCL19, extrahepatic cholangiocarcinoma, epithelial–mesenchymal transition, tumor budding

## Abstract

**Simple Summary:**

Extrahepatic cholangiocarcinoma (EHCC) is an aggressive tumor. The five-year survival rate for patients who undergo surgical resection is only 20–40% due to recurrences. Therefore, elucidating the molecular mechanisms underlying invasion and metastasis in EHCC is crucial for developing adjuvant therapy. The epithelial–mesenchymal transition (EMT) contributes to the metastatic cascade in various tumors. C-C chemokine receptor 7 (CCR7) interacts with its ligand, chemokine (C-C motif) ligand 19 (CCL19), to promote EMT. The association between CCR7 expression and clinicopathological features and EMT status was examined via the immunohistochemical staining of tumor sections from 181 patients with perihilar cholangiocarcinoma. This association was then investigated in two EHCC cell lines. CCR7 mediates cell invasion and migration in EHCC by inducing EMT, which was abrogated by a CCR7 antagonist. CCR7 may be a potential target for adjuvant therapy in EHCC.

**Abstract:**

The epithelial–mesenchymal transition (EMT) contributes to the metastatic cascade in various tumors. C-C chemokine receptor 7 (CCR7) interacts with its ligand, chemokine (C-C motif) ligand 19 (CCL19), to promote EMT. However, the association between EMT and CCR7 in extrahepatic cholangiocarcinoma (EHCC) remains unknown. This study aimed to elucidate the prognostic impact of CCR7 expression and its association with clinicopathological features and EMT in EHCC. The association between CCR7 expression and clinicopathological features and EMT status was examined via the immunohistochemical staining of tumor sections from 181 patients with perihilar cholangiocarcinoma. This association was then investigated in TFK-1 and EGI-1 EHCC cell lines. High-grade CCR7 expression was significantly associated with a large number of tumor buds, low E-cadherin expression, and poor overall survival. TFK-1 showed CCR7 expression, and Western blotting revealed E-cadherin downregulation and vimentin upregulation in response to CCL19 treatment. The wound healing and Transwell invasion assays revealed that the activation of CCR7 by CCL19 enhanced the migration and invasion of TFK-1 cells, which were abrogated by a CCR7 antagonist. These results suggest that a high CCR7 expression is associated with an adverse postoperative prognosis via EMT induction and that CCR7 may be a potential target for adjuvant therapy in EHCC.

## 1. Introduction

Extrahepatic cholangiocarcinoma (EHCC) and its subtypes, especially perihilar cholangiocarcinoma (PHCC), are aggressive tumors. PHCC is defined as a cancer that arises predominantly in the lobar extrahepatic bile ducts, distal to the segmental bile ducts, and proximal to the cystic duct [1]. The therapeutic gold standard for EHCC patients is complete tumor resection. However, due to the incidence of local or distant recurrences, the five-year survival rate for patients who underwent surgical resection is only 20–40% [2,3,4]. Moreover, PHCC, in particular, is often anatomically unresectable. Therefore, elucidating the mechanism of invasion and metastasis in EHCC is essential for developing adjuvant therapy for improving postoperative prognosis. 

The epithelial–mesenchymal transition (EMT) is a crucial step in cancer progression [5,6]. It is a critical physiological process for embryonic development with phenotypic alterations, such as the loss of intercellular adhesion and cell polarity as well as acquisition of migratory and invasive abilities [7,8]. A hallmark of EMT is the downregulation of cell adhesion molecules, such as E-cadherin, and the upregulation of mesenchymal molecules, such as vimentin [7,8]. Aberrant EMT activation in malignant tumors can trigger tumor progression and metastasis [8]. However, the mechanism by which EMT occurs in EHCC remains unknown. Chemokines are a class of small molecule proteins that play significant roles in various physiological, pathological, and immunological responses when bound to their unique chemokine receptors on the cell surface. In the tumor microenvironment, chemokines and chemokine receptors play essential roles in tumor proliferation, invasion, and metastasis. C-C chemokine receptor type 7 (CCR7) is primarily expressed in B cells, naïve T cells, memory T cells, and mature dendritic cells and interacts with chemokine (C-C motif) ligand 19 (CCL19) [9] and CCL21 [10,11]. This interaction is vital for lymphocyte trafficking and lymph node homing [12,13]. Müller et al. first described the expression of CCR7 in the tumor microenvironment (in breast cancer) [9]. CCR7 expression in various tumors has been linked to tumor cell proliferation [14], invasion [15], angiogenesis [16], and metastasis [17]. Additionally, CCR7 induces EMT in the cells of breast [18], gastric [19], and pancreatic cancers [20]. Furthermore, CCR7 expression is correlated with poor prognosis in gastric cancer [21], esophageal squamous cell carcinoma [22], pancreatic cancer [23], urinary bladder cancer [24], and renal cell carcinoma [25]. Therefore, CCR7 overexpression in the tumor microenvironment is critical for cancer progression and metastasis. EHCC has a poor prognosis because it is metastasis-prone. To our knowledge, no study has assessed the frequency and impact of CCR7 expression in EHCC, which may be implicated in the molecular mechanisms underlying EMT, invasion, and metastasis. 

The present study aimed to determine the impact of CCR7 expression on the clinicopathological features and postoperative survival of patients with EHCC and the relationship between CCR7 expression and EMT in human EHCC cell lines. 

## 2. Materials and Methods

### 2.1. Patients

This study included patients (*n* = 181) with PHCC who underwent curative surgical resection between January 1995 and December 2016 at the Department of Gastroenterological Surgery II, Hokkaido University Hospital. Patients who received neoadjuvant chemotherapy or radiotherapy were excluded from the study. Clinical and pathological data of the patients were collected. Written informed consent was obtained from the patients and healthy volunteers from whom peripheral blood samples were collected to isolate peripheral blood mononuclear cells (PBMCs). This study was approved by the Hokkaido University Institutional Review Board (No. 018-0137) and conducted in accordance with the guidelines of the institutional review board and the Declaration of Helsinki.

### 2.2. Tissue Preparation and Histopathology

After gross examination, all excised specimens were fixed in 10% buffered formalin and serially sectioned (3–6 mm). Subsequently, formalin-fixed paraffin-embedded blocks were sliced into 4 μm thick sections and stained for microscopic analysis with hematoxylin and eosin (H&E). Pathological primary tumor (pT) and regional lymph node metastasis were diagnosed as defined by the Cancer Staging Manual, 8th edition, American Joint Committee on Cancer (AJCC) [1].

### 2.3. Tissue Microarray

A tissue microarray [26] of all 181 samples was prepared for immunohistochemistry (IHC) staining. We first reviewed the archived H&E slides of all cases and selected a slide to determine two representative tumor areas (invasive front and center of the tumor) and one representative non-neoplastic area of the bile duct as an internal control. Subsequently, a manual tissue microarrayer (JF-4; Sakura Finetek, Tokyo, Japan) was used to inject 2.0 mm diameter needles into the corresponding sites on paraffin blocks. An experienced pathologist (T.M.) examined a section from each microarray stained with H&E, and evaluated the adequacy of tissue sampling and histopathological diagnosis. 

### 2.4. Immunohistochemistry

Tissue sections were deparaffinized with xylene and rehydrated using a graded ethanol series. Heat-induced antigen retrieval was performed in a high-pH antigen retrieval buffer (BenchMark ULTRA; Roche, Basel, Switzerland). Endogenous peroxidase was blocked by incubation at 36 °C with 3% H_2_O_2_ for 4 min. Sections were subsequently labeled using the horseradish-peroxidase-labeled polymer method (Ventana ultraView DAB Universal Kit; Roche) and an automated immunostaining system (BenchMark ULTRA; Roche). Immunostained sections were counterstained with hematoxylin, dehydrated in ethanol, and cleared in xylene. Finally, sections were stained with anti-CCR7 (clone P-007, 1:100,000 dilution; NB Health Laboratory, Sapporo, Japan), anti-E-cadherin (clone 36, RTU, RRID: AB_397580; Ventana/Roche), and anti-vimentin (clone V9, RTU, RRID: AB_306239; Ventana/Roche) mouse monoclonal primary antibodies. 

### 2.5. Assessment of Tumor Buds

Tumor buds, defined as dedifferentiated cancer cells or clusters of fewer than five cancer cells at the invasive front fields [27,28], reflect the ability of cells to migrate, invade, and metastasize. They are believed to undergo EMT and possess a mesenchymal phenotype [29,30,31]. We previously reported that the quantity of tumor buds at the invasive front fields reflects the EMT status and is associated with tumor invasion and metastasis in EHCC patients [32]. The tumor buds were counted in all 181 patients using previously described methods [32] to evaluate the association between their number and CCR7 expression. Using the International Tumor Budding Consensus Conference method, the tumor buds were counted on the H&E-stained slides containing whole tissue sections (Figure 1A,B) [26]. The H&E slide with the most significant tumor budding was selected for each patient. Tumor buds were counted in a single invasive front field at 200× magnification (0.785 mm^2^), and the areas with the highest density of tumor buds were labeled hot spots [26]. 

### 2.6. IHC Scoring in TMA

We evaluated IHC staining based on the intensity and proportion of positive tumor cells in the TMA cores of 181 PHCC specimens and calculated the IHC score, previously reported as the “H score” [33]. The ordinal values for staining intensity are as follows: grade 0 (negative; Figure 1C), +1 (weak; Figure 1D), +2 (moderate; Figure 1E), and +3 (strong; Figure 1F). In CCR7, the staining intensity of tumor cells was regarded as strong (grade + 3) when the staining of the cytoplasm and tumor cell membranes was similar to that of the surrounding lymphocytes. Next, the percentage of each grade of tumor cells in each spot of the TMA core was estimated. Subsequently, the H scores of the samples were calculated as the product of the percentage of stained cells and corresponding staining intensity (0, 1, 2, or 3) as follows: H score = sum of (IHC staining grade × % of tumor cells with each IHC staining grade). For example, when the percentages of tumor cells with IHC staining grades of 0, +1, +2, and +3 on the spot were 10, 30, 35, and 25%, respectively, the H score of that spot was calculated as (0 × 10 + 1 × 30 + 2 × 35 + 3 × 25) = 175. E-cadherin and vimentin staining was assessed as described previously [32]. Briefly, the staining intensities of tumor buds were initially classified into three groups (grade 0 (negative; E-cadherin, Figure 1G; vimentin, Figure 1J), +1 (weak–moderate; E-cadherin, Figure 1H; vimentin, Figure 1K), and + 2 (strong; E-cadherin, Figure 1I; vimentin, Figure 1L)), and the proportion of tumor cells with each grade on each spot of the TMA core was estimated. Subsequently, the H score was determined using the same formula as CCR7. Three researchers (M.O., Y.N., and T.M.), blinded to the clinical information of patients, independently examined and scored each case. Differences in interpretation were resolved by consensus among the three researchers. 

### 2.7. EMT Status in IHC Examination

We examined the association between the IHC staining scores for CCR7 expression in 181 PHCC specimens and EMT status defined by the combination of E-cadherin and vimentin IHC staining scores using TMA. EMT status was defined and each case was divided into three groups: (1) cases in which the epithelial status (Figure 1I,J) was high E-cadherin with low vimentin expression (E-high/V-low); (2) cases in which the intermediate status was low (E-low/V-low) or high (E-high/V-high) expression of E-cadherin and vimentin; and (3) cases in which the mesenchymal status (Figure 1G,L) was low E-cadherin with high vimentin expression (E-low/V-high). We also examined the association between the IHC staining H scores for CCR7 expression and the number of tumor buds. Median values established the boundaries between low and high H scores for E-cadherin and vimentin. 

### 2.8. Cell Culture

Two human EHCC cell lines, TFK-1 and EGI-1, were used in this study. The cell lines were obtained from the RIKEN BioResource Center (RIKEN BRC, Tsukuba, Japan) and the German Collection of Microorganisms and Cell Cultures GmbH (DSMZ, Braunschweig, Germany), respectively. TFK-1 and EGI-1 were cultured in RPMI-1640 and EMEM supplemented with 4 mM L-glutamine and 2× MEM amino acids—both essential and non-essential. Next, 10% (*v*/*v*) fetal bovine serum (FBS), 100 U/mL penicillin, and 100 μg/mL streptomycin were added to each medium. All cell lines were cultured at a density of 0.5–2.0 × 10^6^ cells/mL at 37 °C and 5% CO_2_. Human peripheral blood aspirates were obtained by venipuncture from a healthy volunteer with written informed consent, and peripheral blood mononuclear cells (PBMCs) were separated by Ficoll density gradient centrifugation. 

### 2.9. RNA Extraction and Reverse Transcription PCR

CCR7 mRNA levels were determined using reverse transcription (RT)-PCR. Total RNA was extracted using an RNeasy Plus Mini Kit (QIAGEN GmbH, Hilden, Germany) according to the manufacturer’s instructions. Samples were quantified using a NanoDrop 2000c Spectrophotometer (Thermo Fisher Scientific, Waltham, MA, USA). cDNA was synthesized using the Super Script^TM^ III First-Strand Synthesis System for RT-PCR (Invitrogen Life Technologies, Carlsbad, CA, USA) using 1.0 μg of RNA following the manufacturer’s protocol. RT-PCR was performed using a GeneTouch Thermal Cycler (Hangzhou Bioer Technology, Hangzhou, China). The PCR conditions were as follows: initial denaturation at 94 °C for 2 min, followed by 33 cycles of denaturation at 98 °C for 10 s, annealing at 60 °C for 30 s, and extension at 68 °C for 30 s (CCR7), followed by 25 cycles of denaturation at 98 °C for 10 s and annealing and extension at 68 °C for 30 s (β-actin). The PCR was performed using KOD Plus Ver.2 (TOYOBO, Osaka, Japan) according to the manufacturer’s instructions. Primers were synthesized at Eurofins Genomics (Tokyo, Japan) and were as follows: CCR7 sense, 5′-ACATCGGAGACAACACCACA-3′ and antisense, 5′-CATGCCACTGAAGAAGCTCA-3′; β-actin sense, 5′-CAACCGCGAGAAGATGACCC-3′ and antisense, 5′-GGAACCGCTCATTGCCAATGG-3′. RT-PCR products were visualized with ethidium bromide (0.5 μg/mL) using a ChemiDoc™ XRS Plus System (Bio-Rad Laboratories; Hercules, CA, USA). Quantitative analyses of data were carried out using TaqMan gene expression assays (CCR7 Assay ID: Hs01013469_m1, and ACTB Assay ID: Hs01060665_g1; Applied Biosystems/Thermo Fisher Scientific). The expression level of CCR7 mRNA was calculated using the ratio of CCR7 mRNA to β-actin mRNA. 

### 2.10. Western Blot Analysis

The cells were cultured with or without 100 ng/mL CCL19 for 48 h. The harvested cells were suspended in lysis buffer containing 50 mM Tris-HCl (pH 8.0), 150 mM NaCl, 1% (*v*/*v*) Nonidet P-40 alternative, 0.1% (*w*/*v*) sodium dodecyl sulfate (SDS), 0.5% (*w*/*v*) sodium deoxycholate, and protease inhibitor cocktail (Promega, Madison, WI, USA) and lysed with sonication successively. The total protein was extracted by centrifugation. The concentration of the extracted protein was determined using a TaKaRa BCA protein assay kit (Takara Bio, Kusatsu, Japan) following the manufacturer’s instructions. For immunoblotting, cell lysates were loaded at a protein concentration of 20 µg per well. Proteins were separated by 7.5% sodium dodecyl sulfate-polyacrylamide gel electrophoresis (SDS-PAGE) and transferred to a polyvinylidene fluoride (PVDF) membrane (Immobilon-P Transfer Membrane; Merck Millipore, Darmstadt, Germany). The membrane was blocked with 5% (*w*/*v*) skimmed milk in Tris-buffered saline (50 mM Tris-HCl, 150 mM NaCl, pH 8.0) with Tween 20 (TBS-T). They were then probed with E-cadherin rabbit polyclonal antibody (1:10,000 dilution; Proteintech Group, Chicago, IL, USA) and vimentin rabbit polyclonal antibody (1:2000 dilution; Proteintech Group) for 90 min at 25 °C. After washing with TBS-T, they were incubated with anti-rabbit IgG antibodies conjugated to horseradish peroxidase (1:10,000 dilution; Jackson ImmunoResearch Laboratories, West Grove, PA, USA) for 60 min at 25 °C. For β-actin, the membranes were incubated with anti-β-actin IgG antibodies conjugated to horseradish peroxidase (1:10,000 dilution; Cell Signaling Technology, Danvers, MA, USA). After applying Amersham ECL Prime Western blotting Detection Reagent (GE Healthcare Limited, Buckinghamshire, UK), the protein bands were visualized using a chemiluminescent detection system (ChemiDoc™ XRS Plus System; Bio-Rad Laboratories). Thereafter, membranes were stripped with Western blotting stripping buffer (Takara Bio) and then washed and re-probed with the aforementioned antibodies. Semi-quantitative analyses of data were carried out using Image Lab™ 5.1 (Bio-Rad Laboratories). β-Actin was used as an endogenous loading control. The experiment was repeated thrice. 

### 2.11. Wound Healing Assay

A Culture-Insert 2 well, which is a silicon gasket with two 70-μL wells (Ibidi, Grafelfing, Germany), was placed on a 24-well plate surface. The cells were added into each chamber (3.5 × 10^4^ cells) and incubated until the cells grew to confluence. Subsequently, the Culture-Insert was removed gently, after which the cells were washed twice with PBS and then cultured in 500 μL serum-free medium with or without 100 ng/mL CCL19 (BioLegend, San Diego, CA, USA). Cell migration to close the gap was examined at 0, 24, and 48 h by microscopy. Areas of the initial gap and the part filled by migration were measured on the images, and the rate of migration area was calculated using the following formula: 

Rate of migration area (%) = (area of the part filled by migration)/(area of initial gap). 

The area was evaluated using a high-resolution imaging and analysis system, BZ-9000 (KEYENCE, Osaka, Japan). The experiment was performed in triplicate. 

### 2.12. Cell Proliferation Assay

The cells were seeded at a density of 2000 cells/well in 96-well plates and incubated for 24 and 48 h with or without 100 ng/mL CCL19. Cell proliferation was assessed using a one-step Cell Counting Kit-8 (CCK-8; Dojindo Laboratories, Kumamoto, Japan). Following the manufacturer’s instructions, 10 µL of CCK-8 solution was added to each well and incubated for 3 h. Finally, the absorbance at 450 and 650 nm was recorded using a SpectraMax 190 reader (Molecular Devices, San Jose, CA, USA). The experiment was repeated thrice. 

### 2.13. Migration and Invasion Assay

Transwell^®^ inserts (6.5 mm diameter, 8 μm pores; Corning Inc., Corning, NY, USA) were used to determine cell migration. For the invasion assay, Matrigel^®^ invasion chambers (6.4 mm diameter, 8 μm pores; Corning Inc.) were used. Each cell resuspended in 200 μL of medium with or without 100 ng/mL CCL19 containing 1% FBS was added to the upper compartment of the chamber. In the migration assay, 5.0 × 10^4^ cells of TFK-1 and 1.0 × 10^4^ cells of EGI-1 were seeded. In the invasion assay, 1.0 × 10^5^ TFK-1 and 1.0 × 10^4^ EGI-1 cells were seeded. Afterward, 500 μL of each medium containing 10% FBS was placed into the lower chamber. In the migration assay, when required, each cell was preincubated for 6 h with 15 μg/mL human CCR7 antibody as a neutralizing antibody (clone 150503; R&D Systems, Minneapolis, MN, USA) or mouse monoclonal IgG as an isotype control (clone 20102; R&D Systems). After 48-h incubation, non-migrating cells on the upper surface of the membrane were removed with a cotton swab. The invading cells on the lower surface of the insert were stained with Diff-Quik stain (Sysmex Corporation, Kobe, Japan) and digitally photographed. The tumor cell invasion area was evaluated using a high-resolution imaging and analysis system, BZ-9000 (KEYENCE). Each experiment was performed in triplicate. 

### 2.14. Statistical Analyses

Qualitative variables were analyzed using the chi-square test. Quantitative variables were analyzed using the one-way analysis of variance (ANOVA) to compare the differences between two or more groups. Comparisons between multiple groups were performed using analysis of variance with a post-hoc pairwise *t*-test. Correlations between CCR7 expression and the number of tumor buds were evaluated using Spearman’s rank method. Overall survival (OS) was calculated using the median time between the date of surgery and the date of the last follow-up or death. The cut-off value of the CCR7 staining H score for discriminating postoperative OS was obtained using a recursive partitioning technique. Kaplan–Meier estimates of OS curves were compared using the log-rank test. In the multivariate analysis of OS, the significance of prognostic factors was investigated using Cox proportional hazards models. The significance level was set to *p* < 0.05, and the confidence interval was 95%. Statistical analysis was performed using JMP software for Windows (version 14.0; SAS Institute, Inc., Cary, NC, USA). 

## 3. Results

### 3.1. CCR7 Expression Was Observed in PHCC Tumor Tissues

In the IHC examination (Figure 1), the CCR7 protein was detected in the cytoplasm and membranes of PHCC cells (Figure 1D–F). The medians (range) of the IHC H scores for CCR7, E-cadherin, and vimentin expression were 90 (0–300), 80 (0–200), and 0 (0–200), respectively. Histograms of the H scores for CCR7 expression are shown in Figure 2A. Based on a cut-off value of 170, the H-score of CCR7 expression was classified into two grades: low-grade, 0–169, and high-grade, ≥170. Based on their median values, the cut-off values for low or high H scores for E-cadherin and vimentin were 70 and 10, respectively.

### 3.2. Clinicopathological Features and Overall Survival Were Associated with CCR7 Expression

High-grade CCR7 expression was associated with poor histological differentiation (*p* = 0.003; chi-squared test) and microscopic venous invasion (*p* = 0.036; chi-squared test) (Table 1). The median (range) follow-up time after surgery was 40 (3–231) months. Table 2 summarizes the results of the univariate and multivariate analyses of the effects of clinicopathological factors on postoperative OS in 181 patients with PHCC. Patients with high-grade CCR7 expression exhibited a significantly worse OS than those with low CCR7 expression according to the log-rank test (*p* = 0.011; Figure 2B). Moreover, multivariate analysis identified CCR7 protein expression level (HR, 1.67, 95% CI, 1.10–2.48, *p* = 0.017) and pT and pN classification as the independent prognostic factors for OS in patients with PHCC. 

### 3.3. High CCR7 Expression Was Associated with EMT 

The intensity of CCR7 expression was correlated with the number of tumor buds (*p* < 0.001, *r* = 0. 247; Spearman’s rank method; Figure 2C). The median (range) scores of CCR7 expression in epithelial status (E-high/V-low), intermediate status (E-low/V-low or E-high/V-high), and mesenchymal status (E-low/V-high) were 90 (0–290), 60 (0–250), and 130 (0–300), respectively. CCR7 expression was significantly higher in the mesenchymal state than that in the epithelial or intermediate states (*p* = 0.019, ANOVA; Figure 2D). 

### 3.4. CCR7 Expression Was Observed in TFK-1 Cells

Before investigating the effect of the CCL19/CCR7 axis on the proliferation, migration, and invasion abilities of EHCC cells via EMT, we screened for endogenous CCR7 expression in TFK-1 and EGI-1. CCR7 mRNA was found in TFK-1 cells but not in EGI-1 cells (Figure 3A). 

### 3.5. CCR7/CCL19 Axis Affects the Expression of EMT-Related Proteins 

Western blot analyses were performed to determine the variation of E-cadherin and vimentin protein levels in each cell line following 100 ng/mL CCL19 treatment for 48 h. E-cadherin expression was significantly decreased (*p* = 0.036; Figure 3B), and vimentin expression was significantly increased (*p* = 0.017; Figure 3B) in TFK-1 cells after stimulation with CCL19. On the other hand, EGI-1 showed no significant change (*p* > 0.05; Figure 3B).

### 3.6. CCL19 Promotes Lateral Migration of TFK-1 Cells 

The lateral migration ability and migration ability enhancement of each cell line following CCL19 treatment were investigated using an in vitro wound healing assay. As shown in Figure 4A, TFK-1 and EGI-1 cells showed slight migration. However, CCL19 treatment significantly enhanced the migration ability of TFK-1 cells at 48 h (*p* = 0.035). On the other hand, in the EGI-1 cells, CCL19 treatment did not show any effect (*p* > 0.05). 

### 3.7. CCL19 Did Not Affect Cell Proliferation 

In the wound healing assay, there is a possibility that the gap could be filled not only by cell migration but also by cell proliferation. Therefore, the degree of cell proliferation under the same culture conditions as the wound healing assay was evaluated. No enhancement in cell proliferation was observed with CCL19 treatment (*p* > 0.05; Figure 4B). 

### 3.8. CCL19 Promotes the Migration and Invasion of TFK-1 Cells 

The vertical migration and invasion abilities of each cell line, and enhancements in these abilities following CCL19 treatment, were investigated using Transwell^®^ inserts and Matrigel^®^ invasion chambers. CCL19 treatment significantly enhanced the migration and invasion abilities of TFK-1 cells at 48 h (*p* = 0.016, Figure 5A and *p* < 0.001, Figure 5B, respectively). However, in the EGI-1 cells, CCL19 treatment did not show any effect (*p* > 0.05; Figure 5A,B). 

### 3.9. CCR7 Antagonist Inhibits CCL19-Mediated Migration of TFK-1 Cells

Next, we examined whether the enhanced migration of TFK-1 cells, stimulated upon CCL19 treatment, could be inhibited by using an anti-CCR7 antibody that could neutralize the effect of CCL19. Pretreatment with an anti-CCR7 antibody completely counteracted the CCL19-mediated migration of TFK-1 cells (Figure 5C). As a control, the IgG isotype control did not suppress the enhancement of CCL19-mediated migration.

## 4. Discussion

In the present study, we aimed to investigate the impact of CCR7 expression on the clinicopathological features and postoperative survival in patients with PHCC and the relationship between CCR7 expression and EMT status in human EHCC cell lines. The results of the clinicopathological examination showed that high-grade CCR7 expression was one of the most adverse postoperative prognostic factors in patients who underwent surgical resections for PHCC, and it was associated with a higher number of tumor buds and mesenchymal status (a combination of low E-cadherin and high vimentin expression). These results indicate that high CCR7 expression may be associated with poor OS via EMT in patients with PHCC. 

Therefore, we investigated the association between CCR7 expression and EMT in human EHCC cell lines in vitro. Among the two human EHCC cell lines tested, CCR7 was detected in TFK-1 but not EGI-1. The CCR7/CCL19 interaction significantly enhanced lateral migration ability in the wound healing assay, along with vertical migration and invasion ability in the Transwell migration and invasion assays of TFK-1 cells. Notably, this effect was not observed in EGI-1 cells. Furthermore, neither cell line exhibited enhanced proliferation after the CCL19 treatment, suggesting that the cell filling observed in the wound healing assay was a result of enhanced migration rather than proliferation. Following treatment with CCL19, the epithelial marker E-cadherin was downregulated, whereas the mesenchymal marker vimentin was upregulated in TFK-1 cells. These results suggest that CCL19 induces EMT in human EHCC cells that are positive for CCR7. 

Furthermore, we found that stimulation of TFK-1 cells with CCL19 increased cell migration, and the anti-CCR7 antibody significantly suppressed this increase. These results confirmed that CCR7 activation is responsible for increased migration, suggesting that CCR7 may be an effective therapeutic target. Beatriz et al. [34] revealed the efficacy of the anti-CCR7 antibody treatment against a xenograft human mantle cell lymphoma model in vivo. However, CCR7 is also responsible for the migration of CCR7-expressing cells, such as natural regulatory T cells or semi-mature dendritic cells, which are involved in immune tolerance [35,36]. Anti-CCR7 therapy may also suppress the migration of such physiological CCR7-expressing cells, and its immediate implementation in clinical practice needs to be carefully considered. In this regard, Winter et al. [37] demonstrated that the chronic deficiency of CCR7 could lead to autoimmune diseases. Because certain types of immunodeficiency may be induced with anti-CCR7 therapy, additional research is required to develop alternative strategies, such as targeting CCR7 exclusively in cancer cells or inhibiting CCR7-associated signal transduction pathways. In future clinical applications, anti-CCR7 antibodies may represent an adjunct therapy to enhance the efficacy of existing chemotherapy techniques, similar to anti-HER2 therapy in breast cancer. This may be particularly important if CCR7 expression is high in resected or biopsy specimens, as the combination of anti-CCR7 antibodies with gemcitabine or cisplatin may inhibit recurrence or promote response.

The present study had several limitations. First, in this retrospective study, samples were collected from a single Japanese center. Notably, a meta-analysis reported that the impact of high CCR7 expression on poor OS was more significant in Asian patients than that in Caucasians [38]. Therefore, a larger and more diverse sample must be analyzed to eliminate the risk of bias and validate the conclusions of this study. Second, IHC was utilized to evaluate clinical tumor samples. Although IHC staining has been widely used in several protein expression studies, it has disadvantages, such as disparities in technical procedures and scoring standards among facilities and studies. Therefore, a multi-institutional prospective study is required to confirm these results. Third, this study is focused on CCL19/CCR7 interactions, but another ligand of CCR7, CCL21, is similar to CCL19 and may also be relevant to the topic of this study. Although CCL19 is considered more efficient than CCL21 in activating ERK1/2 [39] related to EMT, the two ligands share only 32% amino acid identity and have ligand bias [40]. Therefore, there is a possibility that further knowledge will be obtained based on experiments involving CCL21. Fourth, the present study did not explore signaling pathways nor conduct in vivo experiments with animal models that provide a tumor microenvironment similar to that of humans. 

## 5. Conclusions

The present study suggests that CCR7 mediates EMT progression, and high CCR7 expression is associated with a poor prognosis owing to EMT. Therefore, in the future, CCR7 could be a potential therapeutic target for PHCC adjuvant therapy. 

## Figures and Tables

**Figure 1 cancers-15-01878-f001:**
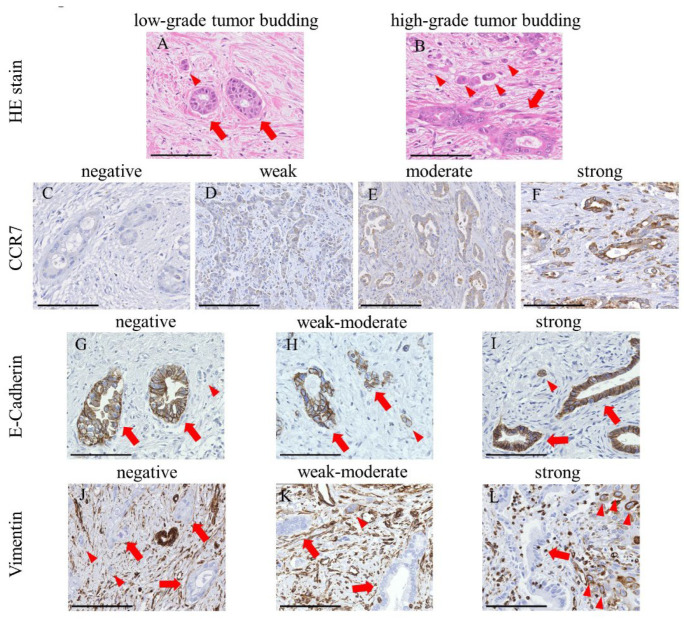
CCR7 expression and epithelial–mesenchymal transition status in perihilar cholangiocarcinoma. (**A**,**B**) Representative images of hematoxylin and eosin staining for low-grade (**A**) and high-grade (**B**) tumor budding from *n* = 181 samples. The arrows and arrowheads indicate neoplastic glandular structures and tumor bud cells, respectively. (**C**–**F**) Immunohistochemical detection of negative (**C**), weak (**D**), moderate (**E**), and strong (**F**) CCR7 expression in tumor cells. CCR7 protein was detected in the cytoplasm and cell membranes of tumor cells and the surrounding lymphocytes. The staining intensity of strongly stained tumor cells was comparable to that of the surrounding lymphocytes. The arrows and arrowheads indicate neoplastic glandular structures and lymphocytes, respectively. (**G**–**I**) E-cadherin was detected in the cell membrane of neoplastic glandular structures, while tumor bud cells showed negative (**G**), weak–moderate (**H**), or strong (**I**) staining for E-cadherin. The arrows and arrowheads indicate neoplastic glandular structures and tumor bud cells, respectively. (**J**–**L**) Vimentin was not expressed in neoplastic glandular structures, while tumor bud cells displayed negative (**J**), weak–moderate (**K**), or strong (**L**) staining for vimentin. The arrows and arrowheads indicate neoplastic glandular structures and tumor bud cells, respectively. Scale bar = 100 μm.

**Figure 2 cancers-15-01878-f002:**
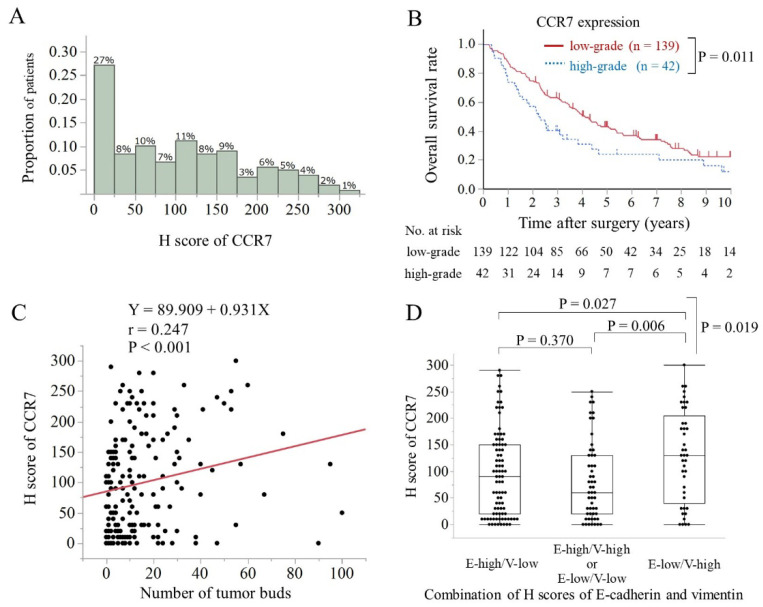
CCR7 expression was associated with poor prognosis and EMT status. (**A**) Histograms displaying patient distribution according to IHC staining scores of CCR7 expression. (**B**) Kaplan–Meier survival curves for the 181 patients with perihilar cholangiocarcinoma based on CCR7 expression grade. According to the log-rank test, patients with high CCR7 expression had a significantly worse long-term overall survival rate than those with low CCR7 expression (*p* = 0.011). (**C**) The correlation between IHC staining scores for CCR7 expression and the number of tumor buds was assessed using Spearman’s rank method. High CCR7 expression grade was associated with the number of tumor buds (*p* < 0.001). (**D**) The relationship between CCR7 expression and the combination of the H scores of E-cadherin (E) and vimentin (V) was assessed using an analysis of variance with post-hoc pairwise *t*-test. High-grade CCR7 expression correlated significantly with EMT status (*p* = 0.019).

**Figure 3 cancers-15-01878-f003:**
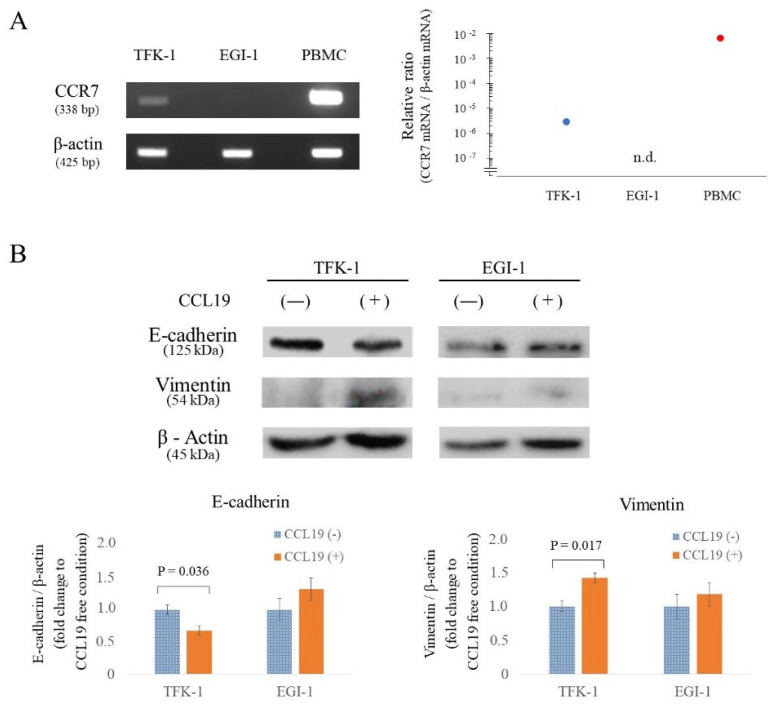
CCR7 expression and the effect of CCL19/CCR7 on EMT-related proteins in cholangiocarcinoma cells. **(A**) RT-PCR analysis of CCR7 mRNA expression in each cell line. CCR7 mRNA levels were standardized with respect to β-actin mRNA. CCR7 mRNA was detected in TFK-1 and PBMCs but not in EGI-1. n.d.; not detected. (**B**) Changes in E-cadherin and vimentin protein levels following treatment with 100 ng/mL CCL19 in each cell line were evaluated using Western blotting. TFK-1 cells showed significant downregulation of E-cadherin (*p* = 0.036) and upregulation of vimentin (*p* = 0.017) in response to CCL19 treatment (one-way ANOVA). Data were normalized to β-actin levels. n.d.; not detected. The uncropped blots are shown in Appendix A.

**Figure 4 cancers-15-01878-f004:**
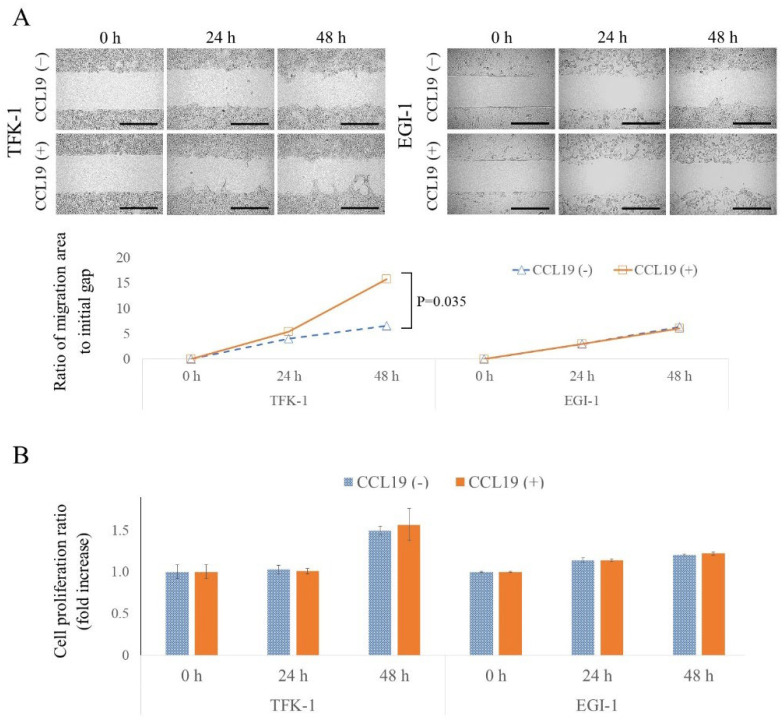
Effect of CCL19/CCR7 on migration and cell proliferation. (**A**) A wound healing assay was used to determine lateral migration ability with or without CCL19 (100 ng/mL) for 24 and 48 h. Migration was significantly enhanced by treatment with CCL19 in TFK-1 cells (*p* = 0.035, one-way ANOVA). Scale bar = 500 μm. (**B**) Cell proliferation assay showing the effect of CCL19 (100 ng/mL) treatment at 0, 24, and 48 h using the WST-8 method. Absorbance at 450 nm was measured against a reference wavelength of 650 nm. Each bar represents the mean ± SD of three independent experiments.

**Figure 5 cancers-15-01878-f005:**
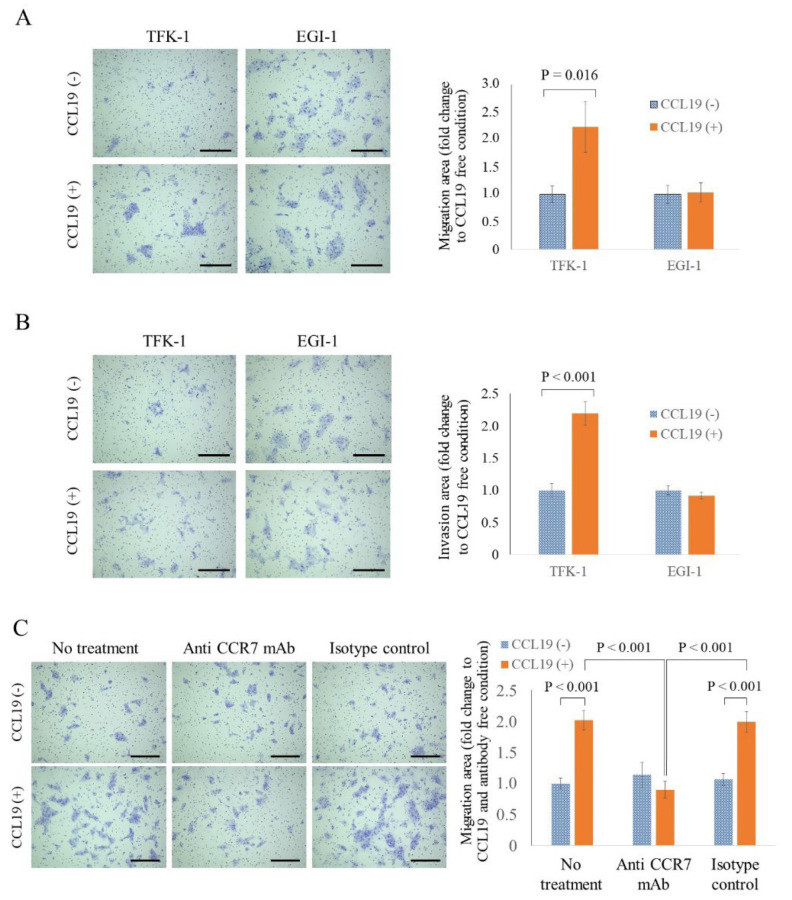
CCR7-dependent CCL19 promoted migration and invasion in cells. (**A**) The migration assay, used to determine the vertical migration ability for 48 h, showed that migration was significantly enhanced by treatment with CCL19 (100 ng/mL) in TFK-1 cells only (*p* = 0.016, one-way ANOVA). (**B**) The invasion assay, used to determine the invasion ability for 48 h, showed that invasion was significantly enhanced by treatment with CCL19 (100 ng/mL) in TFK-1 cells only (*p* < 0.001, one-way ANOVA). (**C**) A migration assay performed after preincubation of TFK-1 with 15 μg/mL anti-CCR7 antibodies for 6 h showed that anti-CCR7 antibody significantly suppressed CCL19-mediated migration (*p* < 0.001, one-way ANOVA). Panels on the left in (**A**–**C**) show representative images of cells at the end-point, while those on the right show the quantification of the area covered by the assayed cells. Scale bar = 100 μm. Each bar represents the mean ± SD of three independent experiments.

**Table 1 cancers-15-01878-t001:** Comparison of the clinicopathological features among patients with different CCR7 expression grades (*n* = 181).

		CCR7 Expression Grades				
		Low-Grade (*n* = 139)		High-Grade (*n* = 42)		
Clinicopathological feature		*n*	(%)	*n*	(%)	*p*
Age	<70	66	(47.5)	27	(64.3)	0.055
	≥70	73	(52.5)	15	(35.7)	
Sex	Male	108	(77.7)	35	(83.3)	0.423
	Female	31	(22.3)	7	(16.7)	
Histological grade	G1	45	(32.4)	4	(9.5)	0.003
	G2	68	(48.9)	23	(54.8)	
	G3	26	(18.7)	15	(35.7)	
pT classification (AJCC, 8th edition)	T1	3	(2.2)	2	(4.8)	0.242
	T2	94	(67.6)	23	(54.8)	
	T3	25	(18.0)	7	(16.7)	
	T4	17	(12.2)	10	(23.8)	
pN classification (AJCC, 8th edition)	N0	75	(54.0)	24	(57.1)	0.674
	N1	55	(39.6)	14	(33.3)	
	N2	9	(6.5)	4	(9.5)	
pM classification (AJCC, 8th edition)	M0	137	(98.6)	41	(97.6)	0.688
	M1	2	(1.4)	1	(2.4)	
Microscopic lymphatic invasion	Absent	66	(47.5)	18	(42.9)	0.598
	Present	73	(52.5)	24	(57.1)	
Microscopic venous invasion	Absent	61	(43.9)	11	(26.2)	0.036
	Present	78	(56.1)	31	(73.8)	
Microscopic perineural invasion	Absent	11	(7.9)	6	(14.3)	0.235
	Present	128	(92.1)	36	(85.7)	
Invasive carcinoma at resected margin	Negative	124	(89.2)	34	(81.0)	0.176
	Positive	15	(10.8)	8	(19.1)	
Median survival time (years)		3.9		2.3		0.018

G1, well differentiated; G2, moderately differentiated; G3, poorly differentiated (according to the AJCC 8th edition). Low-grade, H-score of CCR7 staining 0–169; high-grade, H-score of CCR7 staining ≥170. AJCC, American Joint Committee on Cancer.

**Table 2 cancers-15-01878-t002:** Univariate and multivariate analyses of factors associated with the overall survival of the 181 patients with PHCC.

		No. of Patients		Univariate		Multivariate	
Variable		*n*	(%)	Median Survival (Months)	*p*	Relative Risk (95% CI)	*p*
Age (years)	<70	93	(51.4)	46	0.491		
	≥70	88	(48.6)	44			
Sex	Male	143	(79.0)	46	0.529		
	Female	38	(21.0)	44			
Histological grade	G1	49	(27.1)	56	0.221		
	G2	91	(50.3)	43			
	G3	41	(22.7)	28			
pT classification (AJCC, 8th edition)	T1	5	(2.8)	53	0.001	1	0.026
	T2	117	(64.6)	53		1.07 (0.39–4.45)	
	T3	32	(17.7)	30		1.70 (0.56–7.36)	
	T4	27	(14.9)	21		2.14 (0.71–9.30)	
pN classification (AJCC, 8th edition)	N0	99	(54.7)	76	<0.001	1	<0.001
	N1	69	(38.1)	29		2.32 (1.60–3.36)	
	N2	13	(7.2)	19		3.21 (1.54–6.16)	
pM classification (AJCC, 8th edition)	M0	178	(98.3)	46	0.011	1	0.265
	M1	3	(1.7)	16		2.27 (0.49–7.78)	
Microscopic lymphatic invasion	Absent	84	(46.4)	57	0.094		
	Present	97	(53.6)	30			
Microscopic venous invasion	Absent	72	(39.8)	53	0.009	1	0.183
	Present	109	(60.2)	32		1.30 (0.88–1.93)	
Microscopic perineural invasion	Absent	17	(9.4)	58	0.195		
	Present	164	(90.6)	43			
Invasive carcinoma at resected margin	Negative	158	(87.3)	49	<0.001	1	0.056
	Positive	23	(12.7)	21		1.66 (0.98–2.67)	
CCR7 expression grade	Low-grade	139	(76.8)	50	0.011	1	0.017
	High-grade	42	(23.2)	27		1.67 (1.10–2.48)	

PHCC, perihilar cholangiocarcinoma; G1, well differentiated; G2, moderately differentiated; G3, poorly differentiated (according to the AJCC 8th edition). Low-grade, H-score of CCR7 staining 0–169; high-grade, H-score of CCR7 staining ≥170. AJCC, American Joint Committee on Cancer; CCR7, C-C chemokine receptor type 7; CI, Confidence interval; PHCC, perihilar cholangiocarcinoma.

## Data Availability

The datasets generated and/or analyzed during the current study are not publicly available due to privacy reasons but are available from the corresponding author on reasonable request.

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
