# Peer review of "CCR7 Mediates Cell Invasion and Migration in Extrahepatic Cholangiocarcinoma by Inducing Epithelial–Mesenchymal Transition"

_cancers, 2023, doi:10.3390/cancers15061878_

Round 1

Reviewer 1 Report

Oba et al. studied the mechanism of metastasis in extrahepatic cholangiocarcinoma (EHCC) by analyzing C-C chemokine receptor 7 (CCR7), which interacts with its ligand, chemokine (C-C motif) ligand 19 (CCL19) to promote epithelial-mesenchymal transition (EMT).  The study aims to elucidate the prognostic impact of CCR7 and its potential target for adjuvant therapy.  CCR7 expression, clinicopathological features, and EMT status were analyzed with 181 patients with perihilar cholangiocarcinoma resected specimens, which supported high-grade CCR7 expression associated with poor prognosis.  The authors also demonstrated that CCR7-induced EMT and anti-CCR7 antibodies worked as antagonists in TFK-1 and EGI-1 cell lines.  The results suggest that high CCR7 expression is associated with an adverse postoperative prognosis, and CCR7 may be a potential target for adjuvant therapy in EHCC.

As the authors acknowledged their study’s limitation, the study didn’t explore signaling pathways or conduct in vivo experiments.  However, they proposed a potential target therapy with decent supporting evidence provided by surgical specimens and cell lines.  It is well-written and worthwhile to be published.

In the second paragraph of the discussion, the authors should mention the methods (the wound healing assay for lateral migration ability and proliferation, Transwell invasion assays for vertical migrations and invasion) they used to demonstrate CCL19-induced EMT and the potential effect of antagonist.  These assays were described in the abstract but not in the discussion.  Including the methods will make the article more persuasive to the readers for the results and conclusion.

Author Response

Response:

Thank you for your feedback and suggestions. As you recommended, the following text was added to the second paragraph of the discussion:

"The CCR7/CCL19 interaction significantly enhanced lateral migration ability in the wound healing assay, along with vertical migration and invasion ability in the Transwell migration and invasion assays of TFK-1 cells. Notably, this effect was not observed in EGI-1 cells. Furthermore, neither cell line exhibited enhanced proliferation after the CCL19 treatment, suggesting that the cell filling observed in the wound healing assay was a result of enhanced migration rather than proliferation." (Page 14, Lines 439–444)

Reviewer 2 Report

The authors demonstrate an association between CCR7 and EMT in EHCC. The results provide important insights into understanding the molecular mechanisms that contribute to EHCC invasion and metastasis and could be used for the development of adjuvant therapies. The discovery that CCR7 is involved in EMT in EHCC and can be inhibited by a CCR7 antagonist paves the way for further research into the use of this potential therapeutic target.

According to the 2021 NCCN guidelines, the combination of gemcitabine and cisplatin remains the current first-line chemotherapy regimen for patients with advanced CCA.

Can the Authors discuss how can Anti-CCR7 therapy be implemented in the current CCA therapy?

Author Response

Thank you for bringing up this question. We have added the following to the third paragraph of the discussion: “In future clinical applications, anti-CCR7 antibodies may represent an adjunct therapy to enhance the efficacy of existing chemotherapy techniques, similar to anti-HER2 therapy in breast cancer. This may be particularly important if CCR7 expression is high in resected or biopsy specimens, as the combination of anti-CCR7 antibodies with gemcitabine or cisplatin may inhibit recurrence or promote response.” (Page 14, Lines 461–466)

Reviewer 3 Report

This study elucidated the prognostic impact of CCR7 expression and its association with clinicopathological features and EMT in extrahepatic cholangiocarcinoma (EHCC). The results suggest CCR7 can be a potential target for adjuvant therapy in EHCC. So, it's interesting in EHCC treatment in the future.

Minor comments: CCL21 is another high affiliated ligand for CCR7. Why the author only chooses CCL19 as treatment in this study?

Author Response

Thank you for pointing this out. We have added the following to the fourth paragraph of the discussion: “Third, this study is focused on CCL19/CCR7 interactions, but another ligand of CCR7, CCL21, is similar to CCL19 and may also be relevant to the topic of this study. Although CCL19 is considered more efficient than CCL21 in activating ERK1/2 [1] related to EMT, the two ligands share only 32% amino acid identity and have ligand bias [2]. Therefore, there is a possibility that further knowledge will be obtained based on experiments involving CCL21.” (Page 14, Lines 475-480)

The following references were added as numbers 39 and 40 in the reference list:

  1. Kohout, T.A.; Nicholas, S.L.; Perry, S.J.; Reinhart, G.; Junger, S.; Struthers, R.S. Differential desensitization, receptor phosphorylation, beta-arrestin recruitment, and ERK1/2 activation by the two endogenous ligands for the CC chemokine receptor 7. J Biol Chem 2004, 279, 23214–23222, doi: 10.1074/jbc.M402125200.
  2. Steen, A.; Larsen, O.; Thiele, S.; Rosenkilde, M.M. Biased and G protein-independent signaling of chemokine receptors. Front Immunol 2014, 5, 277, doi: 10.3389/fimmu.2014.00277.